# Pulmonary inflammation promoted by type-2 dendritic cells is a feature of human and murine schistosomiasis

E. L. Houlder [1,2], A. H. Costain[1,2], I. Nambuya[1,3], S. L. Brown[1], J. P. R. Koopman [2], M. C. C. Langenberg[2], J. J. Janse[2], M. A. Hoogerwerf[2], A. J. L. Ridley[1], J. E. Forde-Thomas[4], S. A. P. Colombo[1], B. M. F. Winkel[2], A. A. Galdon[1], K. F. Hoffmann [4], P. C. Cook[1,5], M. Roestenberg [2], H. Mpairwe[3] & A. S. MacDonald [1] ✉

Schistosomiasis is a parasitic disease affecting over 200 million people in multiple organs, including the lungs. Despite this, there is little understanding of pulmonary immune responses during schistosomiasis. Here, we show type-2 dominated lung immune responses in both patent (egg producing) and pre-patent (larval lung migration) murine *Schistosoma mansoni* (*S. mansoni*) infection. Human pre-patent *S. mansoni* infection pulmonary (sputum) samples revealed a mixed type-1/type-2 inflammatory cytokine profile, whilst a case-control study showed no significant pulmonary cytokine changes in endemic patent infection. However, schistosomiasis induced expansion of pulmonary type-2 conventional dendritic cells (cDC2s) in human and murine hosts, at both infection stages. Further, cDC2s were required for type-2 pulmonary inflammation in murine pre-patent or patent infection. These data elevate our fundamental understanding of pulmonary immune responses during schistosomiasis, which may be important for future vaccine design, as well as for understanding links between schistosomiasis and other lung diseases.

*S. mansoni* is a major cause of the neglected tropical disease schistosomiasis, which is characterised by multi-organ morbidity leading to over 200,000 deaths annually[1]. Following mammalian exposure, *S. mansoni* larvae move through the skin to blood vessels, migrating via the lungs before reaching patency in the mesenteric vasculature, where they produce eggs. Pulmonary symptoms can occur in both acute schistosomiasis, encompassing pre-patent larval migratory stages and early patent (egg producing) stages, as well as chronic schistosomiasis in later months and years. In acute schistosomiasis these pulmonary symptoms include cough and shortness of breath, whereas in chronic stages potentially fatal pulmonary hypertension

can develop[2–4]. Schistosome infection can also influence allergic airway conditions, for instance, with inverse correlations observed between *S. mansoni* infection and wheeze[5]. As vaccine-induced protective immune responses may rely upon killing of lung migrating larvae[6–8], a better understanding of the natural immune response to schistosomes in the lung is crucial for informed design of future vaccines.

Current understanding of the immune response to lung migrating schistosome larvae is minimal in murine models, and almost non-existent in humans[9]. In CBA strain mice, larval lung migration has been shown to occur from day 4–21 post-infection, peaking at day 10–12 post-infection before larvae move to the liver[10,11]. Whilst the exact timing of

[1]Lydia Becker Institute of Immunology and Inflammation, University of Manchester, Manchester, UK. [2]Leiden University Center for Infectious Diseases (LU-CID), Leiden University Medical Centre, Leiden, Netherlands. [3]MRC/UVRI and LSHTM Uganda Research Unit, Entebbe, Uganda. [4]Department of Life Sciences, Aberystwyth University, Aberystwyth SY23 3DA, UK. [5]MRC Centre for Medical Mycology, University of Exeter, Exeter, UK. ✉e-mail: Andrew.MacDonald@manchester.ac.uk

larval lung migration in humans is unknown, studies in the olive baboon suggest that this peaks at day 5, with schistosomula moving to the liver by day 9, and with patency and egg production commencing in all systems by around week 5 post-infection[12–14]. Our limited understanding of murine immune responses to lung-migrating larvae is mainly informed by histological studies, which have shown inflammatory foci surrounding lung-migrating larvae[6]. Immune responses in schistosomiasis have been proposed to be initially Th1 dominated, switching to a Th2 response upon egg production[15,16]. More recently, this dogma has been challenged, with the development of low-level Th2 responses observed prior to egg deposition[17–20], including murine thoracic lymph node T cell IL-4 expression observed by flow cytometry during larval lung migration[18]. Whilst illuminating, this study focused on regulatory T cell responses[18], and therefore was limited in scope with respect to inflammatory readouts, with lung tissue or airway inflammatory responses in pre-patent schistosomiasis still undefined.

To our knowledge, no cellular studies have yet been performed to assess the human pulmonary immune response during schistosomiasis, including to larval schistosomula in the first 3 weeks following exposure, when the infection is undiagnosable. After patency and egg deposition (4–9 weeks post-infection) there is some evidence for a systemic type-1 biased inflammatory response in humans, with secretion of IL-1, IL-6 and TNFα from peripheral blood mononuclear cells (PBMCs)[21], though whether this is also the case in the lung is unknown. During chronic patent infection (12 weeks onwards), eggs swept into the lung (likely via portal-systemic shunting) can induce localised type-2 immune responses in both mice and humans[22–24]. Thus, the pulmonary immune responses to pre-patent or patent infection remains poorly understood, in either humans or mice.

Conventional or classical DCs (cDCs) have been shown to be essential to induce type-2 inflammatory responses in the liver and spleen during patent murine schistosomiasis[25,26]. Specifically, murine development of Th2 responses in the small intestine to schistosome eggs, and survival during active schistosome infection, have been shown to require type-2 cDCs (cDC2s)[27,28]. CDC2s represent a subset of cDCs which promote responses to extracellular pathogens such as helminths, in contrast to type 1 cDCs (cDC1s), which drive immunity to intracellular pathogens[29]. Analogous cDC subsets have been described in humans, although their functional roles are less well described[30]. Although changes in total cDCs are observed in human patent *S. haematobium* infection, including reduced toll-like receptor responsiveness and HLA-DR expression[31,32], whether specific cDC subsets are altered during human schistosomiasis is currently unclear.

We have investigated the pulmonary immune response in both pre-patent (lung migrating) and patent *S. mansoni* infection of humans and mice. Using sputum as an accessible proxy for lung responses, we observed a mixed cytokine profile during human pre-patent infection that was not evident during patent infection. However, in both pre-patent and patent human infection, a significant increase in pulmonary DCs, specifically CD1c+ cDC2s was observed. This increase in pulmonary CD11b+ cDC2s was also evident in pre-patent and patent murine infection, where a type-2 dominated immune response was also identified. Using transgenic cDC2-depleted mice, we revealed a requirement for *Irf4*-dependent MGL2+ CD11b+ cDC2s to promote pulmonary type-2 responses during either pre-patent or patent stages of infection. Together, these data increase our fundamental understanding of the pulmonary immune response during pre-patent and patent phases of schistosome infection, in humans and in mice.

## Results
## Pulmonary cytokine and cellular responses are a feature of pre-patent and patent human schistosomiasis

Refined assessment of human lung immune responses during pre-patent lung-stage schistosome infection has not been previously possible. Here, we utilised a pioneering system, in which adult human volunteers were experimentally infected percutaneously with 20 male *S. mansoni* cercariae (Fig. 1A and Supplementary Fig. 1)[19]. Matched sputum samples were obtained from 3 individuals in the week preceding infection (pre), and 11–14 days post-infection (post), to establish the effect of lung migrating schistosome larvae on pulmonary immune responses. Given the low sample size in this data set, no significant changes in sputum cytokines were observed (Fig. 1B). However, even given this limitation, clear trends were observed, with increases in the pro-inflammatory cytokines TNFα and IL-1β, the regulatory cytokine IL-1RA and the chemokines CCL2, CCL17 and CCL3 in the sputum post-infection (Fig. 1B). In contrast, YKL-40 and CCL22 consistently trended to decrease post-infection (Fig. 1B), whereas other cytokines showed no consistent pattern (Supplementary Fig. 7A). Notably, the monocyte chemoattractant CCL2 also tended to increase in serum (Supplementary Fig. 7B), whilst other consistent trends (TNFα, CCL3) were seen only in the sputum. No consistent changes in eosinophils, neutrophils, B cells, CD4+ or CD8+ αβ T cells or γδ T cells were observed post-infection (Fig. 1C and Supplementary Fig. 8). Furthermore, no clear CD4 or CD8 T cell activation, in relation to CD28 or CD127 expression, was observed post-infection (Supplementary Fig. 8). No changes in monocytes were observed (Fig. 1D). However, a significant increase in sputum CD14- CD16- cDCs, and in particular CD1c+ cDC2s, was evident after schistosome infection (Fig. 1D). Taken together, these data revealed that increased production of a range of cytokines and elevated pulmonary CD1c+ cDC2s are previously unappreciated features of the human pulmonary inflammatory response to pre-patent lung migrating schistosome infection.

Next, we wanted to assess whether the pulmonary immune changes observed during pre-patent infection (Fig. 1) would also be evident post-patency during human schistosomiasis. Sputum samples were obtained from *S. mansoni* infected (case) and non-infected (control) individuals from Entebbe, Uganda, an area endemic for schistosome infection (Fig. 2A and Supplementary Fig. 2). There was a range of *S. mansoni* infection intensities in the participants, from light (0–99 eggs/g) to heavy (>400 eggs/g) (Supplementary Fig. 2)[33]. In contrast to the trends observed in pre-patent infection (Fig. 1), patent schistosomiasis was not associated with modified sputum cytokine responses (Fig. 2B and Supplementary Fig. 9A). However, a significant increase in serum IP-10 was observed, in line with a more systemic response expected in patent schistosomiasis (Supplementary Fig. 9B). No significant changes were observed in total sputum cellularity per gram of sputum plugs, or in sputum eosinophils, neutrophils, B cells, CD8+ T cells, CD4+ T cells or γδ T cells (Fig. 2C and Supplementary Fig. 10). No change in CD4+ or CD8+ T cell activation, in relation to CD28 or CD127 expression, was observed (Supplementary Fig. 10). Consistent with pre-patent infection (Fig. 1), a significant increase in sputum CD14-CD16- cDCs, and a strong trend ($p = 0.056$) for an increase in CD1c+ cDC2s, the dominant subset of sputum CD14-CD16− cDCs, was associated with human patent infection (Fig. 2D). Sputum Clec9a+ cDC1s were rare, with Clec9a expressed at low levels in the airways as previously reported (Supplementary Fig. 6)[34]. This suggested that expansion of pulmonary cDCs, and in particular CD1c+ cDC2s, is a feature of human schistosome infection, at multiple stages and in both patent and pre-patent disease.

## Pulmonary type-2 cytokine and cellular responses are a feature of pre-patent and patent murine schistosomiasis

Following our human sample results, we wanted to assess if similar pulmonary inflammation could be observed during murine schistosome infection, to enable mechanistic dissection of this response. Mice were infected percutaneously with 40 or 180 *S. mansoni* cercariae, and pre-patent samples taken at d21 (Fig. 3A). To be comparable to our human sputum studies, we first assessed airway cytokine levels, finding that larval lung migration led to a significant increase in the type-2 associated cytokine Relmα in response to higher dose infection

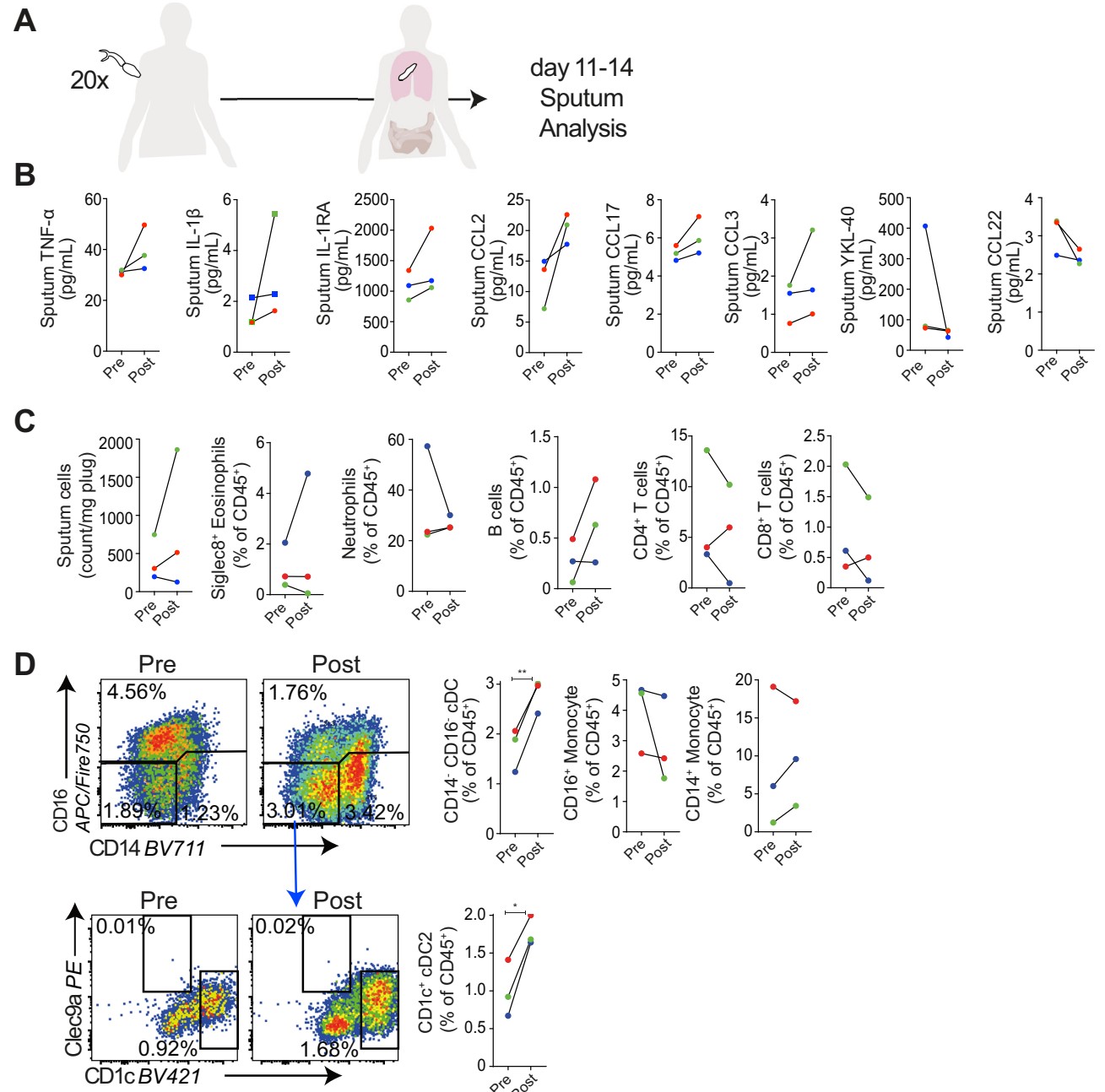

**Fig. 1 | Mixed inflammatory responses, and expansion of sputum cDC2s during pre-patent (lung migratory) schistosome infection of non-endemic participants. A** Pulmonary responses were studied in non-endemic participants in Leiden, The Netherlands infected percutaneously with 20 *S. mansoni* cercariae, with induced sputum samples taken pre-infection, and 11–14 days post-infection. **B** Inflammatory mediators in sputum supernatants were assessed by Luminex and ELISA. **C** Sputum cells were isolated, and assessed by flow cytometry for eosinophils, neutrophils, B cells, CD4+ and CD8+ T cells. **D** Representative flow cytometry plots show changes in monocytes (HLA-DR+, CD14+ or CD16+), cDCs (HLA-DR+, CD14- CD16-), and cDC subsets (CD1c+) following infection. Gate frequencies show % of CD45+ cells. Data are from one study (*n* = 3 individuals), two-sided paired *t* tests were used to compare differences between groups. Colours denote individual participants. \**P* < 0.05, \*\**p* < 0.01. Data are presented as mean values ± SEM. Source data are provided as a Source Data file.

(Fig. 3B)[35]. This type-2 cytokine response was accompanied with an increase of eosinophils collected from the BAL fluid during pre-patent infection at both doses (Fig. 3C). An influx of BAL CD4+ and CD8+ T cells was also observed at the higher dose (Fig. 3C), with differences seen restricted to the airways, and not observed in lung tissue (Supplementary Fig. 11A). Lung CD4+ T cells displayed significantly increased Th1 (IFNγ) and Th2 (IL-4, IL-5, IL-13) cytokine production potential during pre-patent infection at the higher dose (Fig. 3D). Negligible lung CD4+ T cell expression of the regulatory cytokine IL-10 was observed, and there was no lung expansion of regulatory T cells (Tregs) at this

pre-patent timepoint[18] (Fig. 3D, Supplementary Fig. 11B), implicating a mixed Th2/Th1 responses as the hallmark of pre-patent infection in mice infected with either 40 or 180 cercariae.

To understand the pulmonary responses during patent murine schistosome infection, mice were infected with *S. mansoni* and samples taken at d49 (Fig. 4A). We found that patent schistosome infection led to more dramatic airway cytokine changes than pre-patent infection, with a clear trend for an increase in airway levels of Ym-1, Relmα and CCL17 (Fig. 4B). This elevated type-2 inflammatory response was also evident in BAL cellular influx, with numeric and proportional

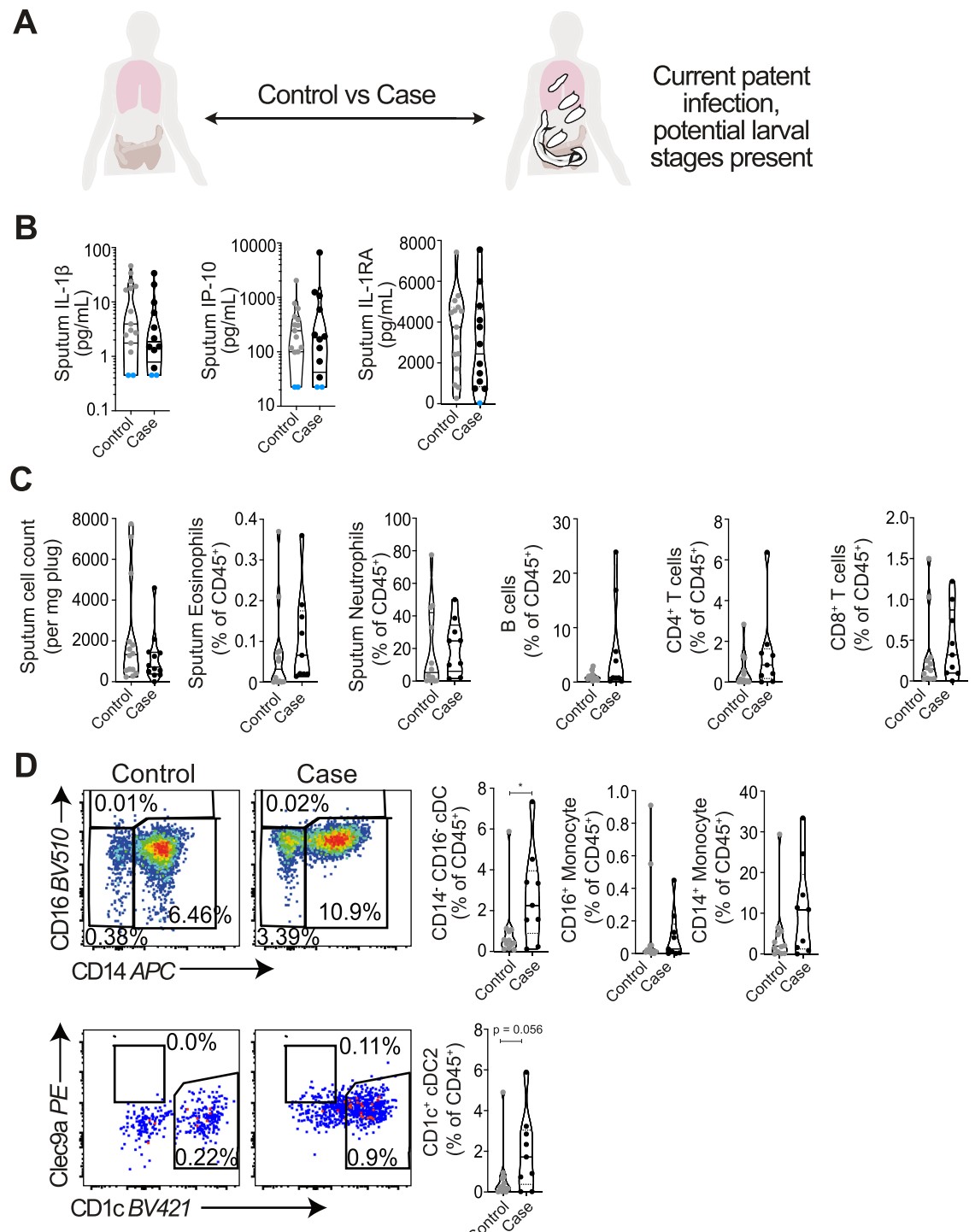

**Fig. 2 | Muted inflammatory responses and expansion of sputum cDCs in patent schistosome infection in endemic participants. A** Pulmonary responses in uninfected control, and *S. mansoni* infected cases from Entebbe, Uganda were compared. **B** Inflammatory mediators in sputum supernatants were assessed by Luminex. Samples below the detection limit were assigned the value of the lowest standard and are shown in blue. Data are from one study (*n* = 27 individuals). **C** Sputum cells were isolated, and assessed by flow cytometry for eosinophils, neutrophils, B cells, CD4⁺ and CD8⁺ T cells. **D** Representative flow cytometry plots show alterations in monocytes (HLA-DR⁺, CD14⁺ or CD16⁺), cDCs (HLA-DR⁺, CD14⁻ CD16⁻), and cDC subsets (cD1c⁺) in infected individuals. Gate frequencies show % of CD45⁺ cells. Data are from one study (*n* = 21 individuals). Two-sided Mann–Whitney tests were used to compare differences between groups. *$p < 0.05$. Data are presented as mean values ± SEM. Source data are provided as a Source Data file.

increase in BAL eosinophils, CD4⁺ and CD8⁺ T cells during patent infection (Fig. 4C), patterns seen to a lesser extent in pre-patent infection (Fig. 3C). The expansion of CD4⁺ and CD8⁺ T cells in the BAL correlated with egg burden in the liver (Supplementary Fig. 21). Cellular responses observed in the airways were mirrored in the lung tissue, where expansions of eosinophils, CD4⁺ and CD8⁺ T cells were observed in patent murine infection (Supplementary Fig. 11A). Further, lung CD4⁺ T cells had significantly increased Th1 (IFNγ) and Th2 (IL-4, IL-5, IL-13) cytokine production potential during patent schistosome infection (Fig. 4D), at a greater magnitude than observed in pre-patent

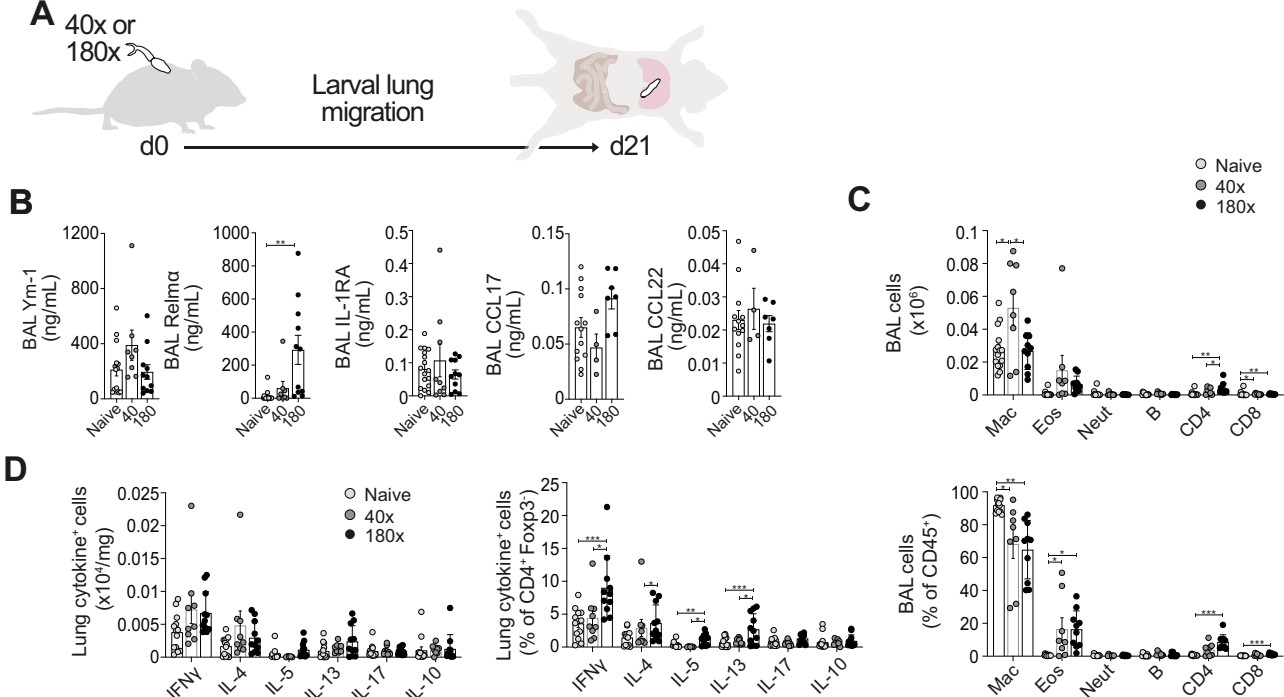

**Fig. 3 | Mixed Th2/Th1 pulmonary responses in pre-patent (lung migratory) murine schistosome infection. A** C57BL/6 mice were percutaneously infected with 40 or 180 cercariae, and samples taken at d21. **B** Inflammatory mediators in BAL were assessed via ELISA. **C** BAL cell isolates were assessed via flow cytometry for macrophages, eosinophils, neutrophils, B cells, CD4+ and CD8+ T cells. **D** Lung cell isolates were stimulated with PMA/ionomycin, and cytokine production assessed via flow cytometry. Data are from 4 independent experiments (*n* = 33 biologically independent animals). Data were fit to a linear mixed effect model, with experimental day as a random effect variable, and groups with a two-sided Tukey's multiple comparison test. *$p < 0.05$, **$p < 0.01$, ***$p < 0.001$. Data are presented as mean values ± SEM. Source data are provided as a Source Data file.

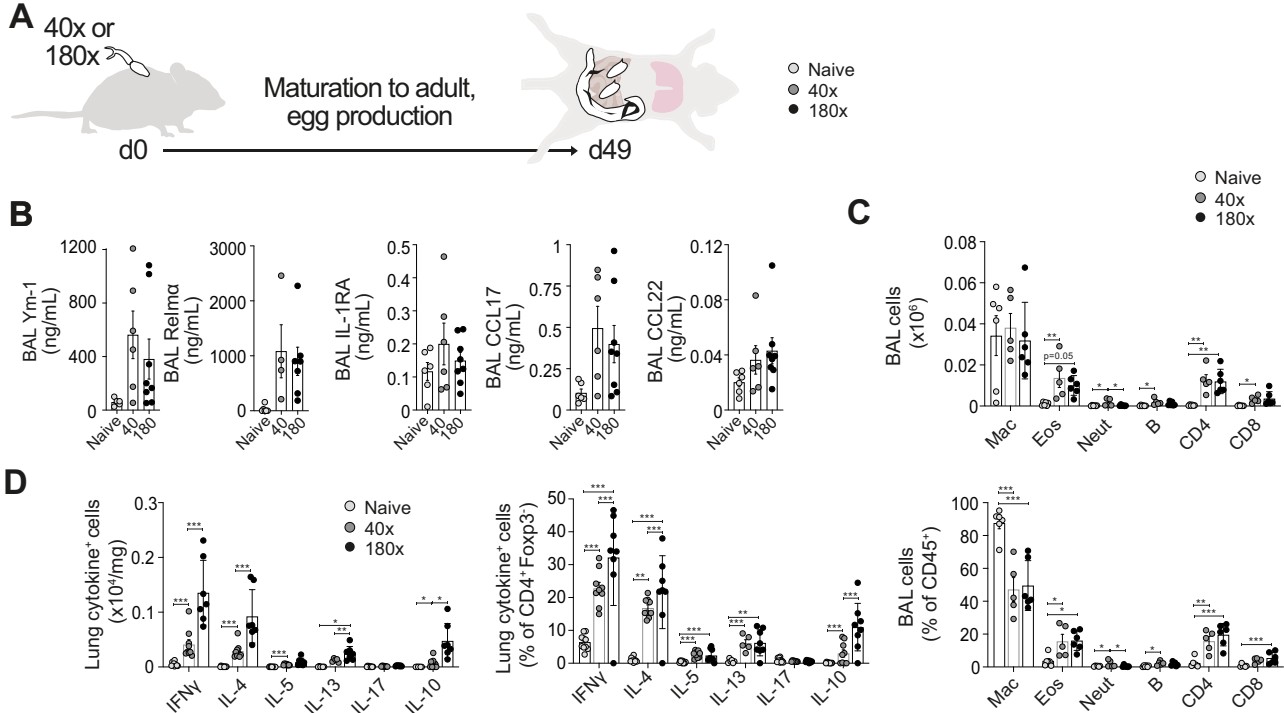

**Fig. 4 | Increased Th2/Th1 pulmonary responses in patent murine schistosome infection. A** C57BL/6 mice were percutaneously infected with 40 or 180 cercariae, and pulmonary samples taken at d49. **B** Inflammatory mediators in BAL were assessed via ELISA. **C** BAL cell isolates were assessed via flow cytometry for macrophages, eosinophils, neutrophils, B cells, CD4+ and CD8+ T cells. **D** Lung cell isolates were stimulated with PMA/ionomycin, and cytokine production assessed via flow cytometry. Data are from 3 independent experiments (*n* = 26 biologically independent animals). Data were fit to a linear mixed effect model, with experimental day as a random effect variable, and groups compared with a two-sided Tukey's multiple comparison test. *$p < 0.05$, **$p < 0.01$, ***$p < 0.001$. Data are presented as mean values ± SEM. Source data are provided as a Source Data file.

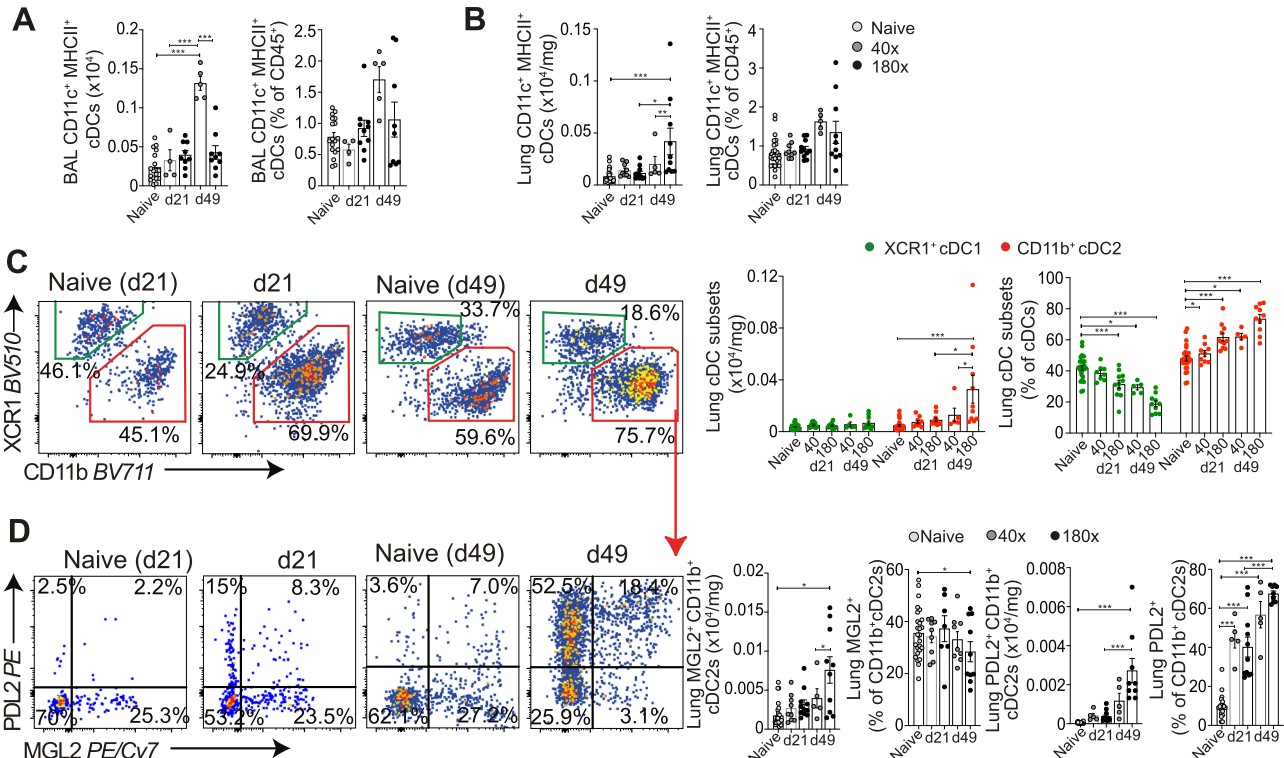

**Fig. 5 | Expansion of pulmonary cDC2s in pre-patent and patent murine schistosome infection.** C57BL/6 mice were percutaneously infected with 40 or 180 cercariae, and samples taken at d21 and d49. Total cDCs (CD11c⁺MHCII⁺) were assessed in (**A**) BAL and (**B**) lung cell isolates. **C** Representative flow cytometry plots show cDC subsets (XCR1⁺ cDC1 and CD11b⁺ cDC2). Gate frequencies show % of MHCII⁺ CD11c⁺ cDCs. **D** Representative flow cytometry plots show changes in cDC2s. Gate frequencies show % of CD11b⁺ cDC2s. Double-positive (MGL2⁺PDL2⁺) cells are counted as both MGL2⁺ and PDL2⁺ in respective graphs. Data are from 7 independent experiments ($n = 61$ biologically independent animals). Data were fit to a linear mixed effect model, with experimental day as a random effect variable, and groups compared with a two-sided Tukey's multiple comparison test. *$p < 0.05$, **$p < 0.01$, ***$p < 0.001$. Data are presented as mean values ± SEM. Source data are provided as a Source Data file.

infection (Fig. 3D). In line with this, there was also a substantial increase in lung CD4⁺ T cells producing the regulatory cytokine IL-10 in patent schistosome infection (Fig. 4D). Although more marked at higher dose infection, similar pulmonary responses were evident at either 40 or 180 cercarial infection (Fig. 4). Looking further, we found an absolute increase, but proportional (% of CD4⁺) decrease in lung Tregs (Supplementary Fig. 11B), suggesting that, whilst a regulatory response was emerging, it was dominated by Th2 inflammation during this relatively early stage of patent infection.

Having identified clear evidence of pulmonary inflammation in pre-patent and patent human and murine schistosome infection (Fig. 1–4), we next wanted to ask if the expansion of cDCs observed in the lung during human pre-patent and patent infection (Figs. 1D, 2D) was also evident in the murine setting at d21 and d49 post-infection. In contrast to human sputum (Figs. 1D, 2D), change in airway (BAL) CD11c⁺MHCII⁺ cDCs was observed only during low dose patent murine schistosome infection (Fig. 5A). At the higher dose, there was an increase in lung tissue CD11c⁺MHCII⁺ cDCs, significant at d49 (Fig. 5B). Schistosome infection induced a dose-dependent expansion of lung CD11c⁺MHCII⁺ DC subsets, in particular CD11b⁺ cDC2s, at d21 and, to a greater extent, at d49 post-infection (Fig. 5C). At both timepoints and doses, these expanded CD11b⁺ cDC2s expressed increased levels of PDL2 (Fig. 5D), which has been associated with Th2 inducing potential of DCs[36]. An absolute increase in CD11b⁺ cDC2s expressing MGL2⁺, a subset required for Th2 induction in other conditions[37], was also observed in response to patent schistosome infection, although they were not proportionally expanded (Fig. 5D). Taken together, these data suggested that murine infection represents some aspects of human pulmonary schistosomiasis, with the conserved expansion of pulmonary cDC2s in both settings during infection.

## cDC2s are required to promote pulmonary type-2 immune responses during murine schistosome infection

To understand the role of CD11b⁺ cDC2s in the lung during schistosome infection, inflammatory responses were studied in mice with a selective deficiency in *Irf4*-dependent cDC2s. First, CD11cΔIrf4 mice[38] were infected with 180 *S. mansoni* cercariae, and samples taken at d21 post-infection (Fig. 6). A significant decrease in *Irf4*-dependent MGL2⁺ CD11b⁺ cDC2s (Fig. 6A) and not XCR1⁺cDC1s, was observed in the lungs of Cre⁺ mice, in line with reports detailing the reduction in intestinal CD11b⁺ and CD103⁺CD11b⁺ cDC2s[27,38] and skin draining lymph node (LN) MLG2⁺PDL2⁺ cDC2s in these mice[36]. This resulted in a significant proportional reduction in BAL eosinophils and a trend for reduced CD4⁺ T cells, along with a significant increase in macrophages in Cre⁺ mice, with no other notable changes (Fig. 6B). These cellular changes were accompanied by a significant reduction in Th2 cytokines IL-4 and IL-5 in Cre⁺ MGL2⁺ CD11b⁺ cDC2 depleted mice, as a proportion of CD4⁺ T cells (Fig. 6C), implicating *Irf4*-dependent dependent cDC2s as critical in promoting type-2 responses during pre-patent schistosomiasis. Notably, whilst efficient depletion of *Irf4*-dependent cDC2s was observed in naïve CD11cΔIrf4 mice, no other significant alterations in pulmonary immune composition were observed (Supplementary Fig. 20).

We next addressed whether *Irf4*-dependent cDC2s were also important for co-ordination of type-2 pulmonary inflammation during patent murine schistosome infection, at d49 (Fig. 7). In line with our results from pre-patent infection (Fig. 6), a deficiency in lung MGL2⁺ CD11b⁺ cDC2s was observed in Cre⁺ CD11cΔIrf4 mice by absolute

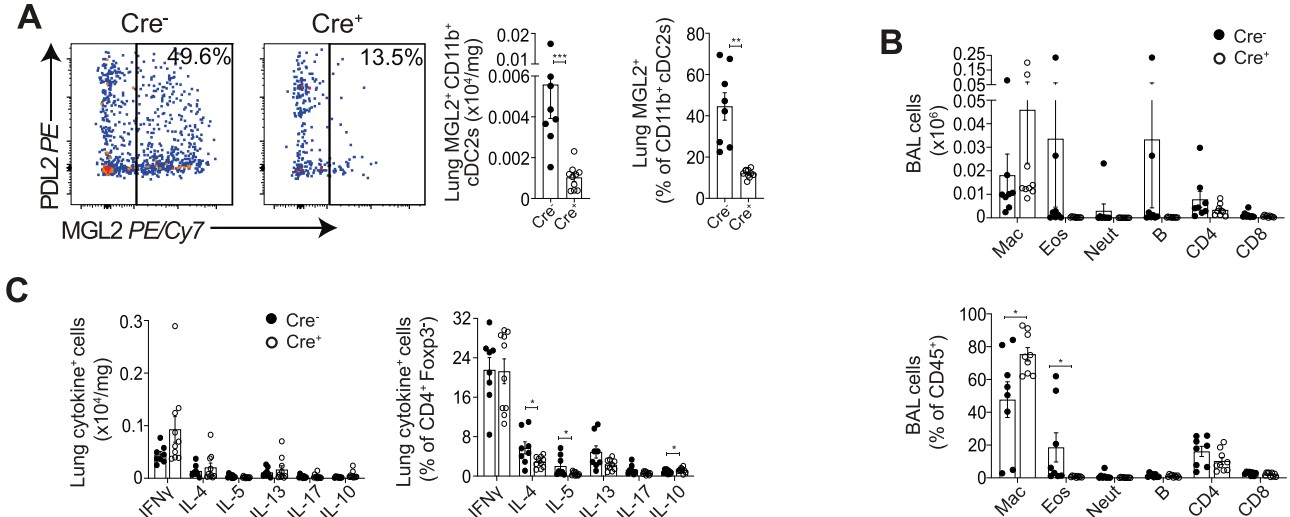

**Fig. 6 | Pulmonary *IRF4*-dependent cDC2s are required to promote pulmonary type-2 immune responses in pre-patent murine schistosome infection.** CD11cΔIrf4 mice were percutaneously infected with 180 cercariae, and samples taken at d21. **A** Representative flow cytometry plots show depletion of lung MGL2⁺ IRF4 dependent cDC2s. Gate frequencies show % of CD11b⁺ cDC2s. **B** BAL cell isolates were assessed via flow cytometry for macrophages, eosinophils, neutrophils, B cells, CD4⁺ and CD8⁺ T cells. **C** Lung cell isolates were stimulated with PMA/ionomycin, and cytokine production assessed via flow cytometry. Data are from 3 independent experiments (*n* = 18 biologically independent animals). Data were fit to a linear mixed effect model, with experimental day as a random effect variable, and groups compared with a two-sided LS means Student's *t* test. *$p < 0.05$, **$p < 0.01$, ***$p < 0.001$. Data are presented as mean values ± SEM. Source data are provided as a Source Data file.

number and as a proportion of total CD11b⁺cDC2s (Fig. 7A). As in pre-patent infection, a significant proportional reduction in BAL eosinophils was observed in Cre⁺ mice, along with a significant proportional and absolute reduction in BAL CD4⁺ T cells by this d49 timepoint (Fig. 7B). Also in line with our results from pre-patent infection, a significant reduction in expression of Th2 cytokines IL-4, IL-5 and IL-13 was observed in cDC2 depleted Cre⁺ mice as a proportion of CD4⁺ T cells, and as absolute number for IL-5 and IL-13 (Fig. 7C). A non-significant trend for increased weight loss was observed in cDC2 depleted Cre⁺ mice (Supplementary Fig. 19), suggesting these cells may protect against morbidity from approximately week 7 post-infection.

Overall, these data strongly suggest that cDC2s, which we have shown to expand in both human and murine pulmonary schistosomiasis, are required to promote lung type-2 inflammation during both pre-patent and patent schistosome infection.

## Discussion

In the current study, we have directly investigated pulmonary responses to *S. mansoni* infection, extending fundamental understanding of this neglected feature of schistosomiasis in both human and murine infection, pre- and post-patency. In either setting, we identified clear evidence of pulmonary inflammation at both stages of infection. Whilst human samples generally showed a muted pulmonary response, more robust type-2 dominated inflammation was observed during murine infection. Strikingly, expansion of pulmonary cDCs, in particular cDC2s, was evident in both human and murine infection in either pre-patent or patent stages of infection. Mechanistically, dissecting this phenotype in CD11cΔIrf4 mice revealed cDC2s to be required for pulmonary type-2 responses in both pre-patent and patent schistosomiasis.

Our human studies reveal for the first-time changes in the pulmonary immune response during pre-patent and patent schistosome infection. During pre-patent infection, several consistent trends were observed in the sputum, including an increase in pro-inflammatory cytokines such as IL-1 and TNFα, warranting further studies of the respiratory response during pre-patent schistosomiasis, with an increased sample size. Notably, this was in line with a recent study which found the production of IL-1α and TNFα was observed upon re-stimulation of PBMCs from *S. mansoni* infected individuals with schistosomula antigens[39]. A more muted immune response was observed in patent human infection, with no significant changes in sputum cytokines perhaps attributable to increased chronicity of infection, with individuals characterised by regular water exposure and infrequent chemotherapeutic treatment. Increased regulatory and reduced T cell proliferative responses have been observed previously in chronically infected individuals, although no prior studies have assessed this in the lung[40,41]. Although use of sputum as a proxy for lung responses is a generally accepted approach[42–44], future work could utilise more invasive tissue sampling (such as lung biopsies) to gain an increased understanding of parenchymal responses to the intravascular schistosome[45].

Our study also provides up-to-date characterisation of the pulmonary immune response during both pre-patent and patent murine *S. mansoni* infection. In the pre-patent phase, our work confirms and expands upon previous observations of increased thoracic lymph node Th2 cells without Treg expansion[18], improving our understanding by revealing increased lung tissue Th2 cytokines, as well as the Th1 cytokine IFNγ. Using BAL as a readout of airway inflammation, we were also able to identify early increases in the type-2 cytokine Relmα, as well as eosinophilia and an influx of T cells. Assessment of airway responses allowed for maximal comparability with our human sputum data, whilst also enabling us to observe subtle immune changes against the low cell number and uniform (>90% macrophage) make up of murine BAL. Interestingly, murine pulmonary immune responses were more noticeable in patent than pre-patent infection, despite negligible expected translocation of strongly Th2-promoting parasite eggs to the lungs at this timepoint[23]. Pulmonary type-2 immune responses during intestinal helminth infection have been previously observed, and proposed to be a protective mechanism against concomitant larval lung migration[46,47]. Alternatively, schistosome induction of type-2 immune responses in the lung could serve to initiate a systemic type-2 response which is permissive for further schistosome development and egg production[17,48,49].

Until now, the dynamics of cDCs in the lung was unknown at any stage of murine or human schistosomiasis. We have revealed a consistent increase in pulmonary cDC2s in both human and murine

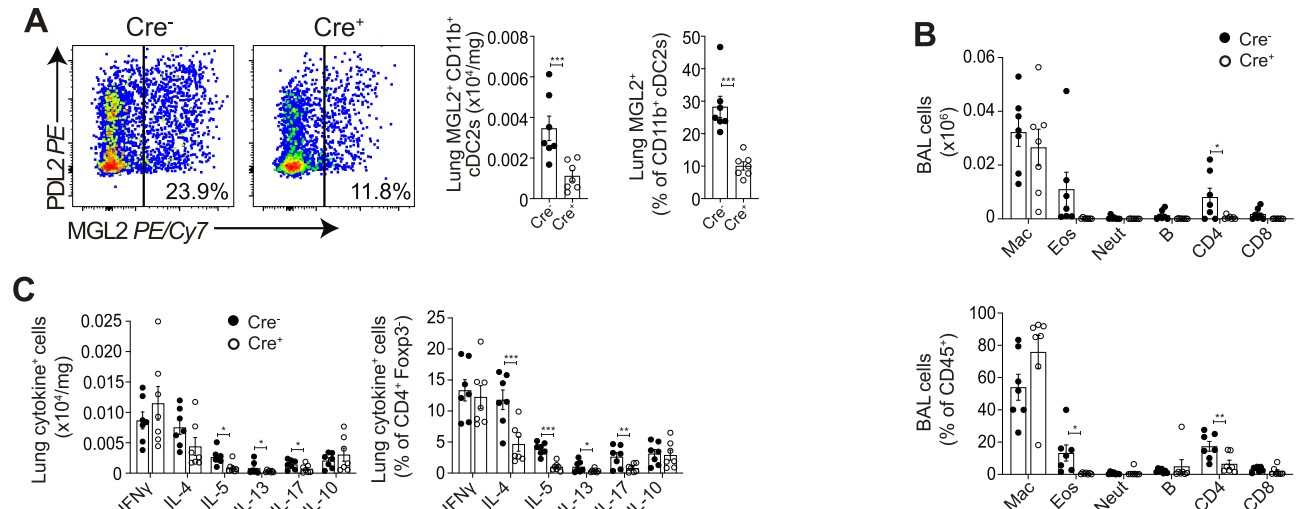

**Fig. 7 | Pulmonary *IRF4*-dependent cDC2s are required to promote pulmonary type-2 immune responses in patent murine schistosome infection.** CD11cΔIrf4 mice were percutaneously infected with 180 cercariae, and pulmonary samples taken at d49. **A** Representative flow cytometry plots show depletion of MGL2⁺ IRF4 dependent cDC2s. Gate frequencies show % of CD11b⁺ cDC2s. **B** BAL cell isolates were assessed via flow cytometry for macrophages, eosinophils, neutrophils, B cells, CD4⁺ and CD8⁺ T cells. **C** Lung cell isolates were stimulated with PMA/ ionomycin, and cytokine production assessed via flow cytometry. Data are from 2 experiments (*n* = 14 biologically independent animals). Data were fit to a linear mixed effect model, with experimental day as a random effect variable, and groups compared with a two-sided LS means Student's *t* test. *$p < 0.05$, **$p < 0.01$, ***$p < 0.001$. Data are presented as mean values ± SEM. Source data are provided as a Source Data file.

settings, at multiple stages of disease. This extends our understanding of the role of cDC2s in murine schistosome infection, building on previous reports of expansion of cDC2s during patent murine schistosome infection in the liver and mLNs[25–27], as well as human studies that have revealed functional alteration of circulating and skin DCs in schistosomiasis[31,32,50,51]. Notably, the continued elevated presence of human cDC2s in the airways suggests they may exert a long-term influence on future pulmonary immune responses, and could therefore critically determine the effectiveness of lung-stage targeted vaccines[52,53] as well as allergic conditions[5].

In addition to characterising DC dynamics, we have utilised the CD11cΔIrf4 mouse model[38] to reveal the functional role of pulmonary cDC2s during schistosomiasis. Specifically, previous studies have shown that cDC2s are required to promote type-2 immune responses in multiple organs, including the lung, in allergic inflammation and helminth infections, including following injection of *S. mansoni* eggs or antigen[27,28,36,54]. In addition, mice lacking *Klf4*-dependent cDC2s show reduced survival by week 7 of *S. mansoni* infection, comparable to infection of IL-4⁻/⁻ mice, implying an inability to induce a type-2 immune response[28]. However, in this previous work, the type-2 immune response was not assessed following cDC2 depletion during active schistosome infection[28]. Therefore, our current study is the first to formally demonstrate a requirement for cDC2s, and specifically *Irf4*-dependent cDC2s, to promote type-2 immune responses during active schistosome infection[38]. This does not imply that *Irf4*-dependent cDC2s are solely responsible for promoting type-2 responses during schistosome infection, with other DC subsets, macrophages or even non-haematopoietic cells potentially contributing during inflammation[55–57].

Previous studies have implicated MGL2 and PDL2 expressed on cDC2s as critical for promoting type-2 responses[27,28,36,37,58]. We now highlight their differential regulation, with PDL2 but not MGL2 expression increasing as a percentage of pulmonary cDC2s during murine *S. mansoni* infection, and the majority of MGL2⁺ cDC2s gaining PDL2 expression in patent infection. In contrast to recently published work suggesting that *Irf4* is required for PDL2 expression[59], we have shown that PDL2⁺ cDC2s are not *Irf4* dependent in the lung, unless they co-express MGL2, with MGL2⁻PDL2⁺ cDC2s remaining at a constant level in CD11cΔIrf4 mice[38]. Together, this could suggest distinct functions of PDL2⁺MGL2⁻, PDL2⁺MGL2⁺ and PDL2⁻MGL2⁺ cDC2s, with the CD11cΔIrf4 mouse model only able to delineate the function of cDC2s that express MGL2[36,38]. The potential for differential roles for MGL2⁺ vs PDL2⁺ cDC2s is supported by recent work showing unique functions of MGL2⁺ cDC2s in lymph node T cell priming[60], as well as numerous studies that have used single-cell sequencing to reveal functional heterogeneity in the cDC2 compartment[61–63]. Together, our work supports the need for further research unpicking the relative roles of *Irf4* dependent (MGL2⁺) and independent (MGL2⁻PDL2⁺) cDC2s during type-2 inflammation, which may be tissue and pathogen specific[27,28].

In summary, our work elevates fundamental understanding of the pulmonary immune response during human and murine pre-patent and patent schistosomiasis. In mice, we have revealed that lung type-2 immune responses are evident from the pre-patent stage, amplifying during patency. In both human and murine settings, we have demonstrated a consistent expansion of pulmonary cDC2s throughout infection. Moreover, we have provided mechanistic insight into the role of these cDC2s, with murine cDC2 depletion revealing their requirement for induction of pulmonary type-2 responses. Our results also provide immunological insight into pulmonary symptoms observed during pre-patent and patent schistosomiasis, which may inform future research aiming to design vaccines for this neglected tropical diseases, as well as help understanding of the broader impact of schistosomiasis on other lung conditions such as allergic airway inflammation[2–5].

## Methods

### Ethics statement

This research complies with all relevant ethical regulations. All procedures performed on mice and invertebrates at Aberystwyth University adhered to the United Kingdom Home Office Animals (Scientific Procedures) Act of 1986 (project license P3BC46FD) as well as the European Union Animals Directive 2010/63/EU and were approved by AU's Animal Welfare and Ethical Review Body (AWERB). All procedures performed on mice in the University of Manchester were ethically reviewed, performed under license and in accordance with the UK Home Office ASPA 1986 and the GSK Policy on the Care, Welfare and Treatment of Animals. Mice were housed in individually ventilated cages, with cage temperature at 23 °C, humidity at 54% and a

12 h light/dark cycle. Biomphalaria glabrata was used for the production of cercariae. These were kept in aquaria at 26 °C, in Lepple water with a 12 hr light/dark cycle.

Pre-patent lung migratory stages of schistosomiasis were studied in a subset of volunteers involved in an experimental human *S. mansoni* infection study (ClinicalTrials.gov Identifier: NCT02755324), carried out at Leiden University Medical Centre (LUMC), as previously described[19], after review and approval by the LUMC Institutional Medical Ethical Research Committee. Informed consent was obtained. The endemic patent infection study was reviewed and approved by the UVRI Research Ethics Committee, as well as the Uganda National Council for Science and Technology and the University of Manchester Research Ethics Committee. Analysis of clinical trial samples from LUMC and UVRI at the University of Manchester was reviewed and approved by the University of Manchester Research Ethics Committee. All human biological samples were sourced ethically and their research use was in accord with the terms of the informed consents under an IRB/EC approved protocol.

### Parasite life cycle

Female Tuck Ordinary (HsdOla:TO; Envigo) mice were maintained at Aberystwyth University (AU) for the purposes of perpetuating the NMRI (Puerto Rican) strain of *S. mansoni*. Mice were percutaneously infected with 180 *S. mansoni* cercariae (~45 min) and were perfused 7 weeks later. At this time, *S. mansoni* eggs were harvested from infected livers by mechanical disruption; bright light-induced miracidia were obtained from liver eggs and used to infect *Biomphalaria glabrata* (NMRI albino and pigmented hybrid lines) snails (6-8 miracidia/snail)[64]. Once patent (~4-5 weeks post-infection), snails were maintained in constant darkness but exposed to light (1 h) in order to prompt the synchronous release of cercariae.

### Murine models

C57BL/6 (C57BL/6JOlaHsd; Envigo) and *Itgax^cre^Irf4^fl^* (referred to as Cd11cΔIrf4) mice were maintained under specific pathogen-free conditions at the University of Manchester[38,65,66]. Male or female mice aged 6–21 weeks were used for analysis. Although samples were not blinded or randomised, infected and non-infected experimental mice were co-housed in order to reduce confounding "cage effects".

Experimental mice were percutaneously infected with 40 or 180 *S. mansoni* cercariae. For Cd11cΔIrf4 experiments, Cre+ mice were excluded when no evidence of depletion of *Irf4*-dependent MGL2+ cDC2s was observed. Similarly, Cre− littermates were excluded from analysis if <1% of their BAL cells were eosinophils at d21 post-infection, as these were not representative of wild-type responses previously observed.

### Human study design

In the pre-patent study, participants were percutaneously exposed to 20 male cercariae, with induced sputum taken from three individuals aged 18–35 in the week preceding infection, and then 11–14 days post-infection (Supplementary Fig. 1). To study endemic patent infection, participants aged 18–25 were recruited from a 2 km² area surrounding the Kigungu Landing site, Entebbe, Wakiso District, Uganda (Supplementary Fig. 2). For these endemic participants, schistosome infection was assayed by Kato Katz and circulating cathodic Ag (CCA) testing (Rapid Medical Diagnostics), and individuals with a previous history of pulmonary disease or current hookworm infection were excluded.

### Cell isolation (murine)

Single-cell suspensions were prepared using the following methods: BAL was collected by washing the peritoneal cavity or lungs with PBS containing 2% FBS and 2 mM EDTA (Sigma). Lungs were processed via incubation at 37 °C with 0.8 U/ml Liberase TL (Sigma) and 80 U/ml DNase I type IV (Sigma) in HBSS (Gibco), as previously described[67]. After 40 min the digestion was halted with PBS containing 2% FBS and 2 mM

EDTA, and suspensions passed through 70 µm cell strainers. Red blood cells (RBCs) were lysed using RBC lysis buffer (Sigma) and leukocytes then used for flow cytometry or stimulation. To assess cytokine secretion potential, lung cells were stimulated for 3 h at 37 °C with 30 ng/ml PMA (Sigma), 0.5 µg/ml ionomycin (Sigma) and 1 µl/ml GolgiStop (BD) in X-vivo-15 (Lonza), supplemented with 1% ʟ-glutamine (Gibco) and 0.1% β-mercaptoethanol (Sigma). Stimulations were carried out in a 96 well u bottom plate, at $4 \times 10^5$ cells per well in 200 µl final volume.

### Cell isolation (human)

Sputum was induced via oral administration of 200 µg Salbutamol, followed by inhalation of nebulised 4.5% NaCl for up to 4 min, after which sputum expectoration was attempted[68]. NaCl inhalation and attempted sputum expectoration was repeated until a sample was produced, up to 3 times within 30 min. Forced expiratory volume was monitored throughout the procedure, and sputum placed on ice prior to processing. Sputum plugs were isolated, weighed and a 4x volume of 0.1% dithiothreitol (Sigma) was added and shaken at room temperature for 15 min[68]. An equal volume of PBS was then added, and samples filtered through sequential 100 then 40 µm cell strainers. Samples were centrifuged and supernatants and cells cryopreserved at −80 °C, with cells resuspended in freezing media (50% RPMI (Gibco) with 50% FCS (Sigma) and 10% DMSO (Sigma)) controlled freezing of sputum cells was obtained by use of a Mr Frosty freezing container (Nalgene).

### Flow cytometry (murine and human)

For murine lung tissue, $1 \times 10^6$ cells per sample were stained, while for murine BAL and human sputum the entire sample was stained. Samples were washed with PBS and stained for viability with ZombieUV or ZombieNIR (1:2,000; Biolegend). Samples were then blocked with 5 µg/ml αCD16/CD32 (2.4G2; BioLegend) or Human FcBlock (BD) in FACS buffer (PBS containing 2% FBS and 2 mM EDTA) before staining for surface markers at 4 °C for 30 min. After staining, cells were washed twice in FACS buffer and then fixed in 1% paraformaldehyde in PBS for 10 min at room temperature. For detection of intracellular proteins, cells were fixed with BD cytofix/cytoperm (BD) for 1 h, then washed three times with 1x eBioscience permeabilization buffer (Thermofisher) and antibodies to intracellular markers added for overnight staining. Samples were acquired on BD Fortessa or LSRII flow cytometers with FacsDiva (BD) software and analysed with Flowjo v10 (Tree Star). Gating schemes, representative flow cytometry plots and selected FMOs for human and murine cell populations are provided in Supplementary Figs. 3–6 and 12–18. For human sputum analysis, sentinel gating[69] was used to look at a numerous cell types in a 16-17 colour flow cytometry panel. Specifically, certain sentinel markers (CD3, CD16, CD45, CD11c, and CD66b) were used to specify cell populations, for instance CD3 for T cells, and were assigned a unique fluorophore and channel. Other markers with non-overlapping expression shared channels, for instance CD8 and CD19 both utilised the BB700 fluorophore, with CD8-BB700 used to gate CD3+ cells as CD8+ T cells, and CD19-BB700 to gate CD3- cells as B cells. The other markers which shared fluorophores were CD4 and CD11b, CD28 and CD1c, TCRβ and Siglec 8, TCRγδ and CD14 or HLA-DR (study dependent) and CD127 and Clec9a. To account for variable quality of sputum, a total of 6 endemic patent samples (3 case, 3 control) were excluded from flow cytometric analysis samples when the parent population (CD3+, HLA-DR+ or CD66b+ cells) had less than 100 cell events.

Antibodies used in murine experiments were as follows: B220 (clone RA3-6B, BV650, 1:100, BioLegend, 103241), CD11b (clone M1/70, BV711, 1:1000, BioLegend, 101242), CD11c (clone N418, BV605, 1:600, BioLegend, 117333), CD19 (clone eBio1D3 (1D3), APC/e780, 1:100, eBioscience, 47-0193-82), CD16/32 (Fcblock) (clone 2.462, Purified, 1:200, BD Biosciences, 553141), CD3 (clone 17A2, APC/e780, 1:100, eBioscience, 47-0032-82), CD4 (clone RM4-5, AF700, 1:200, BioLegend, 100536), CD45 (clone A20, BV785, 1:800, BioLegend, 110743),

CD45 (clone A20, PE, 1:800, BioLegend, 110707), CD64 (clone X54-5171, BV421, 1:100, BioLegend, 139309), CD8 (clone 53-6.7, BV785, 1:200, BioLegend, 100750), Foxp3 (clone FJK-16s, ef450, 1:200, eBioscience, 48-5773-82), IFN(clone XMG1.2, BV711, 1:200, BioLegend, 505835), IL-10 (clone JESS-16E3, BV605, 1:200, BioLegend, 505031), IL-13 (clone W17010B, APC, 1:200, eBioscience, 159405), IL-17 (clone TC11-18H10, PE Cy7, 1:200, BioLegend, 506922), IL-4 (clone 11611, PE Dazzle 594, 1:200, BioLegend, 504132), IL-5 (clone TRFK5, PE, 1:200, BioLegend, 504304), Ly6G (clone 1A8, APC/Cy7, 1:200, BioLegend, 127624), Ly6G (clone 1A8, PerCP Cy5.5, 1:200, BioLegend, 127624), MerTK (clone 2BioI42, APC, 1:100, BioLegend, 151508), MGL2 (clone URA-1, PE Cy7, 1:400, BioLegend, 146808), MHCII (clone M5/114.15.2, PE Cy5, 1:20000, BioLegend, 107612), NK1.1 (clone PK136, APC/e780, 1:200, eBioscience, 47-5941-82), NK1.1 (clone PK136, PE Cy5, 1:200, BioLegend, 108716), PDCA-1 (clone 927, BV650, 1:200, BioLegend, 127019), PDL2 (clone TY25, PE, 1:200, BioLegend, 107206), SiglecF (clone E50-2440, PE CF594, 1:400, BD Biosciences, 562757), TCR (clone H57-597, APC/e780, 1:100, eBioscience, 47-5961-82), Ter119 (clone TER-119, APC/e780, 1:200, eBioscience, 47-5921-82) and XCR1(clone 2ET, BV510, 1:400, BioLegend, 107206).

Antibodies used in human experiments were as follows: CD3 (clone UCHT1, BB515, 1:2000, BD Biosciences, 564465), CD8 (clone RPA-T8, BB700, 1:2000, BD Biosciences, 566452), CD19 (clone H1B19, BB700, 1:100, BD Biosciences, 745907), CCR3 (clone 5E8, APC, 1:200, BioLegend, 310708), CD4 (clone RPA-T4, AF700, 1:200, BioLegend, 300526), CD11b (clone ICRF44, AF700, 1:200, BioLegend, 301355), CD16 (clone 368, APC/Fire750, 1:200, BioLegend, 980114), CD28 (clone CD28.2, BV421, 1:200, BioLegend, 102127), CD1c (clone L161, BV421, 1:200, BioLegend, 331526), CD45 (clone H130, BB515, 1:200, BD Biosciences, 564585), TCR (clone IP26, BV605, 1:200, BioLegend, 306732), Siglec8 (clone 837535, BV605, 1:200, BD Biosciences, 747872), CD11c (clone BU1S, BV650, 1:50, BioLegend, 337238), TCRgd (clone 11F2, BV711, 1:50, BD Biosciences, 568490), CD14 (clone M5E2, BV711, 1:200, BioLegend, 301837), HLA-DR (clone L243, BV785, 1:200, BioLegend, 307642), CD127 (clone A019DS, PE, 1:200, BioLegend, 986002), Clec9a (clone 8F9, PE, 1:100, BioLegend, 353803), CD66b (clone G1OFS, PE CF594, 1:200, BioLegend, 305121), CD7 (clone CD7-667, PE Cy5, 1:100, BioLegend, 343110), Foxp3 (clone 236 A/E7, PE Cy7, 1:200,eBioscience, 25-4777-42), CD1a (clone H1149, PE Cy7, 1:200, BioLegend, 300122), CD45 (clone H130, BV785, 1:100, BioLegend, 304048), HLA-DR (clone L243, BV711, 1:100, BioLegend, 307644), CD16 (clone 3G8, BV510, 1:100, BioLegend, 302048) and CD14 (clone M5E2, APC, 1:50, BioLegend, 982506).

### Soluble cytokine analysis (murine and human)
ELISAs and Luminex were used to assess soluble cytokines in sputum supernatants and BAL. R&D DuoSet ELISA kits were used to analyse murine Ym-1, Relmα, IL-1RA, CCL17 and CCL22, and human YKL-40, CCL17 and CCL22, according to the manufacturer's instructions. Plates were read at 450 nm, with 570 nm as the reference wavelength, using an Infinite M200 Pro plate reader (Tecan). ELISA data was analysed using Prism v9 (GraphPad), with a sigmoidal dose–response curve fitted to log-transformed standards to interpolate samples. In human sputum supernatants and serum a Bio-Plex Pro Human Cytokine 27-plex Assay (BioRad) was performed according to the manufacturer's instructions. Raw data were analysed on BioPlex Manager (Biolegend), and a weighted 5 parameter logistic regression curve fitted to standards to interpolate data. Analytes in which over two thirds of the samples had values extrapolated below the range of the standard curve were not included. Analytes in which less than two thirds of the samples were below the range of the standard curve were included, with samples below the range of the standard curve assigned the value of the lowest standard. By doing this we were able to retain the information provided by a below-range sample, whilst ensuring improper conclusions drawn from extrapolation below the standard range were not presented[70]. For sputum supernatants from human pre-patent

lung migratory infection, the following cytokines were below detection limit: IL-2, IL-4, IL-5, IL-7, IL-9, IL-10, IL-12p70, IL-13, IL-15, IL-17, Eotaxin, bFGF, PDGF-BB, VEGF, IFNγ, G-CSF and GM-CSF. For sputum supernatants from human endemic patent infection, the following cytokines were below detection limit: IL-2, IL-4, IL-5, IL-7, IL-9, IL-10, IL-12p70, IL-13, IL-15, IL-17, MCP-1, MIP-1α, MIP-1β, Eotaxin, bFGF G-CSF, GM-CSF, PDGF-BB, VEGF, RANTES and TNFα. Cytokines presented from serum were chosen to match those that were presented in the corresponding sputum analyses. For serum from human endemic patent infection the following cytokines were below the detection limit: IL-1β, IL-6, IFNγ, CXCL8. For pre-patent endemic human serum, samples were sent to Olink for assessment via their proteomic platform at Uppsala, Sweden, using proximity extension assays (PEA) and quantitative real-time polymerase chain reaction (qPCR). Protein levels reported are relative Normalised Protein Expression (NPX), which have been linear scaled ($2^{NPX}$) to Arbitrary Units (AU). Serum levels of the following cytokines were not measured in pre-patent infection: IL-1β, IL-1RA, CCL17, YKL-40, CCL22 and CXCL8.

### Statistical analysis
For murine experiments, mixed linear models were fitted utilising Rstudio (R Core Team). To account for random variation, the experimental day was designated a random effect variable, with timepoint or genotype as fixed effect variables. To compare multiple groups a post-hoc Tukeys HSD test was used. For the analysis of human data, Prism (GraphPad Software) v7-9 was used. For statistical analysis of responses in human pre-patent lung migrating infection, it was not possible to test normality due to the low n number, and therefore normality was assumed, and $t$ tests used. For statistical analysis of responses in human patent infection, a D'Agostino-Pearson normality test was performed, with samples found to be non-normally distributed and so a non-parametric test (Mann-Whitney) was used. No correction was made for multiple comparisons. A small sample size was chosen for this exploratory work, with the individual organism the experimental unit in all analyses. All tests performed were two-sided.

### Reporting summary
Further information on research design is available in the Nature Portfolio Reporting Summary linked to this article.

## Data availability
Source data underlying figures and Supplementary Figures are provided with this paper in the Source Data file. Source data are provided with this paper.

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

## Acknowledgements

The authors thank the participants in the Netherlands and Uganda for taking part in this study. We appreciate the work done by the field and clinical teams at MRC/UVRI and LSHTM Uganda Research Unit, and at LUMC, as well as the guidance of Alison Elliott and Maria Yazdanbakhsh to facilitate this study. We thank members of the Lydia Becker Institute and MacDonald laboratory (University of Manchester) for scientific discussions and some experimental assistance, and the University of Manchester and UVRI and flow cytometry facilities. We thank Matthew Edwards, Semra Kitchen and Alexander Phythian-Adams (GSK) for scientific discussions. William Agace provided the *Itgax^cre Irf4^fl* mice. This work was supported by a BBSRC CASE studentship (with GSK) to ELH (BB/P504543/1), MCCIR core, GCRF IAA, HIC-Vac and MRC funding to ASM (MR/W018748/1). K.F.H. and J.F.T. were supported by the Wellcome trust (107475/Z/15/Z). M.R. was supported by a Veni grant (no. 016.156.076) from the Netherlands Organization for Health Research and Development and a Gisela Thier Fellowship (no. 14-0645) from LUMC. Part of the work was conducted at the MRC/UVRI and LSHTM Uganda Research Unit which is jointly funded by the UK Medical Research Council (MRC) part of UK Research and Innovation (UKRI) and the UK Foreign, Commonwealth and Development Office (FCDO) under the MRC/FCDO Concordat agreement and is also part of the EDCTP2 programme supported by the European Union.

## Author contributions

A.S.M., E.L.H., H.M. and M.R. were responsible for conceptualisation. E.L.H., A.H.C., I.N., J.P.R.K., S.L.B., M.C.C.L., J.J.J., M.A.H., A.J.L.R., J.E.F.-T., B.M.F.W., A.A.G., P.C.C., S.A.C., conducted or enabled investigations. K.F.H., J.E.F.-T., A.S.M., M.R. and H.M. provided resources. E.L.H. and A.S.M. wrote the manuscript with input from all other authors. A.S.M., H.M., and M.R. were involved in funding acquisition.

## Competing interests

The Manchester Collaborative Centre for Inflammation Research is a joint venture between the University of Manchester and GSK. The authors working at the MCCIR (A.S.M., E.L.H., A.H.C., S.L.B., A.J.L.R., P.C.C., A.A.G., S.A.C.) declare that the research was conducted in the absence of any commercial or financial relationships that could be construed as a potential conflict of interest. The remaining authors declare no competing interests.
