## [Peer Review File · Nature Communications]

Pulmonary inflammation is a feature of human and murine schistosomiasis, promoted by type-2 dendritic cellsREVIEWER COMMENTS

Reviewer #1 (Remarks to the Author):

The manuscript by Houlder et al reports a series of human and murine investigations into the lung phenotype of acute and chronic schistosomiasis infection, and the role of a specific subset of dendritic cells (Irf4+ cDC2s) in the Type 2 inflammation in the lungs. Overall the studies employ an elegant combination of controlled human infection, chronic human samples, and mechanistic murine studies. There is specific study of the acute phase of infection in which schistosomulae migrate through the lungs to the target organ (portal venous system), which is an important part of the life cycle but has not been well studied previously.

1. Related to the human samples, there is particular emphasis on sputum analysis which may have an uncertain relationship to intraparenchymal inflammation as would be induced by an intravenously-located parasite. This should be mentioned at least as a limitation. Can the authors comment on how a cDC2 stimulated by an intravenous antigen would end up in the airspace? Does this require destruction of tissue integrity allowing leakage of cells, or would a cDC2 naturally migrate into the lungs?
2. It is noted that the cytokine signal was stronger and more consistent in the mouse model than in the human samples: this is probably related to the cercariae dose (20 cercariae/70kg human, vs 180 cercariae/20g mouse). In the humans, there may have been substantial heterogeneity in regions of lung that are affected vs not affected. In the chronically infected mice, there is some heterogeneity in the burden of eggs which end up in the lungs following embolization via portocaval shunts. For example, see figures 1C and 6B in Crosby et al 2009 (PMID: 19965814). Can you correlate the degree of inflammation with the burden of eggs, such as using a 4% KOH digest (see Cheever 1968; PMID: 4881073)? This heterogeneity will also be a limitation of the cohort from Uganda; was there a correlation seen between cytokine/cell numbers with stool egg burden, which could plausibly correlate with liver disease severity/egg burden in the lungs?
3. The increase in macrophages in the Irf4-fl;CD11c-Cre mice suggests a potential compensatory shift in antigen presenting cells in the absence of cDC2s. Similarly, other non-traditional antigen presenting cells may present antigen in settings of inflammation, such as endothelial cells (ie, PMID 29312357) or fibroblasts (ie, PMID 35029648), and could also be upregulated when cDC2s are deleted. The roles of these cells may differ between acute and chronic infection, and may explain in part why some of the effects on inflammation by deleting cDC2s was only partial. Can you assess the potential for antigen presentation by some of these other candidates?
4. Fig 5: a substantial portion of the cDC2s are double-positive for PDL2 and MGL2: there is discussion of the function of these proteins in isolation but should be mentioned to be present on the same cells in combination (presumably the DP cells are exceptionally capable of stimulating CD4 cells to Th2); in the quantification shown, do these cells count as both MGL2+ and PDL2+ in the respective graphs?
5. Can the authors show that the cDC2s are actually picking up and/or functionally presenting schistosome antigen?
6. Minor: legend in figure 7 is missing.
7. Minor: Figures 6 and 7: please clarify which graphs belong to panel B (ie. the right-most 2), and in the legend change reference from panels D to C.

Reviewer #2 (Remarks to the Author):

The authors set out to determine the role of CDCs in the lung immune response to schistosome larval migration. While the question is important and interesting, there are some serious flaws with the experimental design and flow analysis that adversely affect the conclusions. Major issues are outlined below

- 1) The sample size for the human prepatent analysis of 3 individuals (2 males 1 female) makes the data uninterpretable. There are known biological sex differences in lung immunity to both infectious and autoimmune/allergic diseases (well described for ILCs in humans (PMCID

PMC5860819 and many others) and in the supplementary figures it is clear that one of the three samples looks very different from the others beginning at the FSC vs SSC level. With this defect, this data does not add anything meaningful to the manuscript, and in fact raises a lot of questions.

- 2) The flow strategy used for the human samples of putting two antibodies into the same fluorophore is very questionable, and based on the limited flow plots shown in the supplementary figures, clearly has led to compensation issues (CD4 for example).
- 3) The human Clec9a antibody does not seem to stain very well in most of the samples, therefore all conclusions based on this gating are questionable.
- 4) The fact that very few flow plots are shown (and none for the intracellular staining) is very concerning based on the poor compensation seen in the human gating strategy. All types of data used to generate conclusion needs to be shown (in concatenated plots) in the paper, not just graphs of numbers
- 5) More minor concern, the human controlled experiments are done with a very low cercariae dose (20) and the endemic data includes majority low to moderate infection intensities. It therefore does not make sense that the mouse experiments were done at a high dose (150) it is quite possible that inflammation from migrating cercariae can have a dose effect, at the very least the mouse experiment should be repeated at a low dose (50-70 cercariae for B6 background mice).
- 6) The majority of the murine data in Figures 3-5 are intriguing, but still have some issues (Why are the gates different for days 21 and 49, there is a significant difference in the space between the gate at the two timepoints).
- 7) Figure 6 and 7 are the strongest of the paper, with the main weakness being no demonstration that loss of the prepatent DC response has an adverse effect on wither morbidity or mortality. What do these cells do for the host?
- 8) Minor point, the murine experiments with 7 mice per group and 2 experiments using both males and females is very likely underpowered to observe a sex-difference (no power calculations for including sex as a biological variable are included in the methods), but the authors strongly state that there are no sex differences.

Reviewer #3 (Remarks to the Author):

Overview

This is a detailed and thorough study of dendritic cell responses to the helminth parasite *Schistosoma mansoni*, including a first analysis of controlled infection in human volunteers, naturally infected endemic country residents, and laboratory mice. Taken together the authors make a strong case for the cDC2 subset being expanded and functionally required for the Type 2/Th2 inflammatory response that ensues in the lung.

While technically and logistically impressive, the manuscript is largely descriptive and confirms earlier work on cDC2s rather than reveals any new major aspect. Nevertheless, this is a high quality and important contribution to the schistosomiasis field.

Detailed comments

1. The Introduction and relevant Results section should more clearly define the DC subsets and the markers now being used to delineate them. Figure Legends should also include this information rather than simply "cDC" or "cDC2" on the y axes, because markers used by different laboratories evolve over time.
2. The Introduction does not mention what I thought was a long-standing paradigm in the field, that an initial Th1 response to schistosome infection switches to Th2 only once egg production has started; does the current manuscript now overturn that notion? If so, it may be worth giving the issue greater prominence.
3. The authors state in the Introduction "vaccine-induced protective immune responses rely upon killing of lung migrating larvae", quoting a study in mice. Surely vaccines could also target other stages of the parasite infection, in skin or bloodstream?
4. While the authors correctly claim that earlier work has not well characterized the pulmonary immune response, in this study there is little comparison with systemic (eg spleen cells in mice,

serum cytokines in patients) that earlier reports presented. I feel that the authors' argument that the lung is the key locus of the response requires more direct comparability with other sites in the same individual cases.

5. After Figures 1 and 2 on controlled and natural human infection respectively, the authors show mouse model data from days 21 (lung stage) and 49 (patent stage). It would be preferable to show day 21 and 49 in the same figure (as is done for Fig 5) so that increases/decreases over time are more readily appraised.

6. Although the CD11c Δ Irf4 mice are an elegant model to demonstrate cDC2 requirements for Th2 responses in schistosomiasis, a more direct test would be to transfer DCs of different genetic and immune status to either WT or gene-targeted mice, for example to rescue the response in the Δ Irf4 mice. This would also exclude off-target effects of the transgenic constructs on other populations (for example, it is notable that the Δ Irf4 mice show a complete loss of pulmonary B cells).

7. Figure 6 A shows MGL2+ cells from WT and gene targeted mice, by implication in both cases infected; it is important to also show naive uninfected data, as the baseline levels may differ at steady state, and the degree of expansion in each genotype should be evaluated.

20th December 2022

Dear Reviewers,

We are pleased to submit a revised version of our manuscript entitled “**Pulmonary inflammation is a feature of human and murine schistosomiasis, promoted by type-2 dendritic cells**” for your consideration.

We thank the reviewers for their time in considering our work and for their supportive comments and suggestions, which we feel have improved the depth and clarity of our manuscript. Below is a point-by-point response to all queries raised, indicating where new data and text has been added to the manuscript. We have made several modifications to the newly revised manuscript to address these suggestions, including the addition of substantial new *in vivo* data (additional data to Figs. 1, 2, 3, 4, 5 and eight new supplementary Figs.), highlighting any text changes in red.

Yours sincerely,

Andrew S. MacDonald
Professor of Immunology

Reviewer 1

Related to the human samples, there is particular emphasis on sputum analysis which may have an uncertain relationship to intraparenchymal inflammation as would be induced by an intravenously-located parasite. This should be mentioned at least as a limitation.

- *We agree that use of sputum as a proxy for lung responses, though a generally accepted approach, is a potential limitation of our study. To acknowledge this, we have added new text to the manuscript Discussion (p13), highlighting that tissue sampling (such as lung biopsies) would be an important next step for this work. Unfortunately, tissue sampling was not feasible as part of the human studies presented in our current manuscript.*

Can the authors comment on how a cDC2 stimulated by an intravenous antigen would end up in the airspace? Does this require destruction of tissue integrity allowing leakage of cells, or would a cDC2 naturally migrate into the lungs?

- *Prior work has shown DCs to be present both intravascularly and in airways in the human lung in both healthy and diseased states¹, with recruitment of DCs into the airspaces in acute pulmonary challenge (LPS inhalation)⁴⁻⁷². Live imaging of murine lungs has revealed migration of DCs within the lung parenchyma, and extension dendrites into the airspaces³. Taking this previous work into account, we would speculate that, upon recognition of a pathogen or damage associated molecular patterns, cDC2s can be expected to be able to migrate into the airspaces without a requirement for destruction of tissue integrity. Schistosome larvae, which are approximately 18µm wide when entering the lung, must bypass thin walled and 6µm wide lung capillaries⁴. It is possible that the resultant mechanical stress could cause the release of damage signals on both sides of the alveolar/ endothelial monolayer, which could mediate cDC2 activation and recruitment into the airspaces.*

It is noted that the cytokine signal was stronger and more consistent in the mouse model than in the human samples: this is probably related to the cercariae dose (20 cercariae/70kg human, vs 180 cercariae/20g mouse).

- *To address this important point, we have added substantial additional murine data to the revised manuscript, directly comparing pulmonary responses in 40 vs 180 cercarial infections at d21 and d49 (revised Figures 3,4 & 5). We have also added new text (p8-9) to highlight that, although more marked at higher dose infection, similar pulmonary responses were evident at either 40 or 180 cercarial infection intensity.*

In the humans, there may have been substantial heterogeneity in regions of lung that are affected vs not affected.

- *We agree that this is an interesting possibility, but would require tissue sampling to address (as discussed above).*

In the chronically infected mice, there is some heterogeneity in the burden of eggs which end up in the lungs following embolization via portocaval shunts. For example, see figures 1C and 6B in Crosby et al 2009 (PMID: 19965814). Can you correlate the degree of inflammation with the burden of eggs, such as using a 4% KOH digest (see Cheever 1968; PMID: 4881073)?

- *In order to isolate enough cells for detailed immune profiling, we were unable to spare lung tissue to digest for egg counts in our experiments. However, as part of the immune cell isolation process, we did not observe lung eggs at the timepoints we studied (d21 and d49). This would be in line with Figure 1E of Crosby et al 2009 (PMID: 19965814), in which very few (a mean of only 2) eggs per lung lobe were identified by week 7 post infection. In the absence of lung egg data, we performed correlations (Spearman's rank) comparing liver egg counts (per gram tissue) from the murine model (week 7) to BAL cell subsets (eosinophils, neutrophils, macrophages, B cells, CD4⁺ T cells and CD8⁺ T cells, as a % of CD45⁺ cells). There was a significant correlation between BAL CD4⁺ and CD8⁺ T cells and liver egg counts (respectively, $r = 0.49$ and 0.63 , $p=0.047$ and 0.0084). If the reviewer feels that this is a key point, we would be happy to include these data as a supplementary Figure in the revised manuscript.*

This heterogeneity will also be a limitation of the cohort from Uganda; was there a correlation seen between cytokine/cell numbers with stool egg burden, which could plausibly correlate with liver disease severity/egg burden in the lungs?

- *No significant correlations were found between stool egg burden and sputum cellular inflammation or cytokines in the human endemic study. If the reviewer feels that this is a key point, we would be happy to include these data as a supplementary Figure in the revised manuscript.*

The increase in macrophages in the *Irf4-fl;CD11c-Cre* mice suggests a potential compensatory shift in antigen presenting cells in the absence of cDC2s. Similarly, other non-traditional antigen presenting cells may present antigen in settings of inflammation, such as endothelial cells (ie, PMID 29312357) or fibroblasts (ie, PMID 35029648), and could also be upregulated when cDC2s are deleted. The roles of these cells may differ between acute and chronic infection and may explain in part why some of the effects on inflammation by deleting cDC2s was only partial. Can you assess the potential for antigen presentation by some of these other candidates?

- *Since BAL macrophage absolute cell numbers were not significantly affected in the *CD11cΔIrf4 Cre⁺* mice, we would suggest that the proportional increase in BAL macrophages observed (Fig. 7) can likely be attributed to the decrease in other BAL cell populations (eosinophils, CD4⁺ T cells) in the *Cre⁻* mice, rather than a compensatory shift in antigen presenting cells, when cDC2s were depleted. Our proposal that cDC2s appear to be primarily responsible for promoting inflammation during lung stage schistosome infection is supported by previous literature showing that cDC2s are required to promote type-2 immune responses in the lung and other tissue sites, in either allergic inflammation or during helminth infection (including following injection of *S. mansoni* eggs or antigens)⁵⁻⁸. However, we completely agree that other cell types (including macrophages) may play a complementary role in antigen presentation, and this may explain in part why the effect of depleting cDC2s was only partial in our experiments. We have added new text to the Discussion to help highlight this important point (p14). We would suggest that further experiments to assess the role of other antigen-presenting cells in this setting would be complex and time consuming, and outwith the scope of the current manuscript.*

Fig 5: a substantial portion of the cDC2s are double-positive for PDL2 and MGL2: there is discussion of the function of these proteins in isolation but should be mentioned to be present on the same cells in combination (presumably the DP

cells are exceptionally capable of stimulating CD4 cells to Th2); in the quantification shown, do these cells count as both MGL2⁺ and PDL2⁺ in the respective graphs?

- *We thank the reviewer for raising this interesting point, and have altered the Discussion to more explicitly highlight these DP cells (p15). We have also added text to the Legend of Fig. 5 to clarify that double-positive (MGL2⁺PDL2⁺) cells were counted as both MGL2⁺ and PDL2⁺ in respective graphs.*

Can the authors show that the cDC2s are actually picking up and/or functionally presenting schistosome antigen?

- *To understand if cDC2s are picking up or presenting antigen from larvae in the lung or lung draining lymph nodes during active infection would be extremely challenging, particularly since schistosomes stably expressing fluorescent proteins are not yet available. While we eagerly await such transgenic parasites, their development currently remains a major technical challenge for the field, and beyond the scope of the current manuscript. We have previously demonstrated the ability of cDC2s to present labelled schistosome antigen (schistosome egg antigen, SEA) in the mesenteric lymph node following direct SEA injection into the small intestine serosa⁵, supporting the fundamental ability of mucosal cDC2s to perform this function.*

Minor: legend in figure 7 is missing.

- *Apologies for this error, which we have corrected.*

Minor: Figures 6 and 7: please clarify which graphs belong to panel B (ie. the right-most 2), and in the legend change reference from panels D to C.

- *Apologies for this error, which we have corrected. Panel B graphs have been moved right to increase the separation from panels A and C, which we hope increases clarity.*

Reviewer 2

The sample size for the human prepatent analysis of 3 individuals (2 males 1 female) makes the data uninterpretable. There are known biological sex differences in lung immunity to both infectious and autoimmune/allergic diseases (well described for ILCs in humans (PMCID PMC5860819 and many others) and in the supplementary figures it is clear that one of the three samples looks very different from the others beginning at the FSC vs SSC level. With this defect, this data does not add anything meaningful to the manuscript, and in fact raises a lot of questions.

- *We agree that the sample size for the human prepatent analysis is small, and with 3 individuals it is not possible to correct for the effects of biological sex. Despite this, we were able to observe a consistent and statistically significant increase in cDC2s in these samples, attesting to the reliability of this phenotype and the power of longitudinal (pre vs post infection) sampling of human donors. This increase in cDC2s was also evident in cross-sectional sampling of endemic donors (n=9 infected vs 12 uninfected controls). We would suggest that this consistency in outcome, even with such small sample sizes, shows that these key data regarding*

cDC2s (the cellular focus of our manuscript) are both remarkable and meaningful. Indeed, both other reviewers complemented the human infection components of our study, while publication of in-depth immunological and clinical results from smaller (n=1) sample sets is common practice in clinical case reports⁹.

The flow strategy used for the human samples of putting two antibodies into the same fluorophore is very questionable, and based on the limited flow plots shown in the supplementary figures, clearly has led to compensation issues (CD4 for example).

- *While we would have preferred to use a more familiar staining and analysis strategy, numerous factors necessitated the use of a sentinel gating approach for flow cytometry assessment of our human samples. Firstly, the low cell yield from sputum meant that it was not possible to use multiple flow cytometry panels. Secondly, the novelty of this research meant that we could not predict which (if any) cell populations would be altered in sputum by infection, necessitating as broad a panel as possible. Finally, conducting the flow cytometry on-site in Uganda (where only a BD LSRII flow cytometer was available) was an ethical requirement for that work. Use of the same sentinel gating approach for the Leiden pre-patent samples was then important for consistency with and comparability to the Ugandan data. Although neither of the other Reviewers raised sentinel gating as an issue (with Reviewer 3 commending our work as ‘detailed and thorough’, ‘technically and logistically impressive’ and ‘high quality and important’), we have added an additional supplementary Figure (Supp. Fig. 12), with concatenated sputum plots and fluorescence controls, which we hope convinces of the reliability of this approach. Although a slight compensation issue was observed in the CD4 channel, we hope that it is clear from the plots that this did not affect our ability to distinguish CD4⁺ from CD8⁺ T cells.*

The human Clec9a antibody does not seem to stain very well in most of the samples, therefore all conclusions based on this gating are questionable.

- *In Supp. Fig. 6 we have shown Clec9a staining of sputum and matched blood samples (stained, acquired and analysed in parallel). We observed convincing Clec9a staining in matched blood samples and therefore concluded that the antibody was working appropriately. In line with our data, low numbers of cDC1s in airway samples have been observed previously¹. However, we appreciate that discussion of alterations in such low numbers of sputum cDC1s could be over-interpretation of our data, and have therefore removed Discussion text, as well as removing panels quantifying cDC1 frequencies from Figs 1 and 2.*

The fact that very few flow plots are shown (and none for the intracellular staining) is very concerning based on the poor compensation seen in the human gating strategy. All types of data used to generate conclusion needs to be shown (in concatenated plots) in the paper, not just graphs of numbers.

- *We have added extensive supplementary data to address these concerns (Supp. Figs 13-18), which provide representative gating schemes for populations identified in both murine and human experiments, and concatenated plots for populations shown in the main Figures.*

More minor concern, the human controlled experiments are done with a very low cercariae dose (20) and the endemic data includes majority low to moderate

infection intensities. It therefore does not make sense that the mouse experiments were done at a high dose (150) it is quite possible that inflammation from migrating cercariae can have a dose effect, at the very least the mouse experiment should be repeated at a low dose (50-70 cercariae for B6 background mice).

- *We agree that this is an important point, and have added substantial additional data to the revised manuscript, highlighting comparable murine pulmonary responses in 40 vs 180 cercarial infections at d21 and d49 (revised Figures 3, 4 and 5).*

The majority of the murine data in Figures 3-5 are intriguing, but still have some issues (Why are the gates different for days 21 and 49, there is a significant difference in the space between the gate at the two timepoints.

- *Apologies for any confusion on this point. The gates are different for days 21 and 49 because mice were infected on the same day (as this is one of the most variable steps in these experiments), and samples were stained and acquired on different days (including a matched naïve sample at each timepoint for comparison). Such day-to-day technical variation is expected (caused by e.g. flow cytometer voltage fluctuations over time), and did not impair our ability to distinguish immune cell types and subsets.*

Figure 6 and 7 are the strongest of the paper, with the main weakness being no demonstration that loss of the prepatent DC response has an adverse effect on either morbidity or mortality. What do these cells do for the host?

- *We did not see a significant impact of cDC2 deficiency on morbidity or mortality in our experiments. Out of a total of 12 Cre⁻ and 11 Cre⁺ infected mice studied up to d49, 2 Cre⁺ animals (18%) had to be euthanised at d43, as they had reached the severity limit of our animal license (20% weight loss). No Cre⁻ mice had to be euthanised. There was a non-significant trend for increased weight loss in Cre⁺ cDC2 depleted mice during week 7 of infection, as is now shown in Supp. Fig. 19 and described on p11 of the revised manuscript. Together, this might suggest a role for cDC2s in limiting host morbidity and mortality from approximately week 7 of infection. Further work would be needed to address this potentially interesting result, beyond the scope of our current manuscript focus on the lung response to infection at d21 and d49.*

Minor point, the murine experiments with 7 mice per group and 2 experiments using both males and females is very likely underpowered to observe a sex-difference (no power calculations for including sex as a biological variable are included in the methods), but the authors strongly state that there are no sex differences.

- *We apologise for over-emphasising this in our initial submission, and have altered the Methods text (p16) to remove any comment on sex differences in our experiments.*

Reviewer 3

The Introduction and relevant Results section should more clearly define the DC subsets and the markers now being used to delineate them. Figure Legends should also include this information rather than simply “cDC” or “cDC2” on the y axes, because markers used by different laboratories evolve over time.

- *As requested, we have added information on key markers used to delineate cDC subsets to the Introduction, Results and Figures of the revised manuscript.*

The Introduction does not mention what I thought was a long-standing paradigm in the field, that an initial Th1 response to schistosome infection switches to Th2 only once egg production has started; does the current manuscript now overturn that notion? If so, it may be worth giving the issue greater prominence.

- *Unfortunately, this is a general misunderstanding in the field that we and others have tried to correct in recent years. We have added text to the Introduction of the revised manuscript (p3-4) to clarify that, although it was initially proposed that immune responses in schistosomiasis are Th1 dominated before egg production^{10,11}, low level Th2 responses are in fact evident prior to egg deposition¹²⁻¹⁴.*

The authors state in the Introduction “vaccine-induced protective immune responses rely upon killing of lung migrating larvae”, quoting a study in mice. Surely vaccines could also target other stages of the parasite infection, in skin or bloodstream?

- *We agree that our wording was a little too strong on this point in our original submission, and have added the word ‘may’ to this sentence (p3) to remove the suggestion that lung migrating larvae are the only stage that can be targeted for vaccination. We have also added additional references (refs 6-8 in main text) on this interesting topic - one providing human evidence for lung responses being important in schistosome vaccination, and a review giving an overview of immune responses that lead to protection in schistosomiasis.*

While the authors correctly claim that earlier work has not well characterized the pulmonary immune response, in this study there is little comparison with systemic (eg spleen cells in mice, serum cytokines in patients) that earlier reports presented. I feel that the authors’ argument that the lung is the key locus of the response requires more direct comparability with other sites in the same individual cases.

- *To address this reasonable point, we have added systemic (serum) cytokine data from our human studies (Supp. Figs 7B&9B). We have not included systemic outputs from murine pre-patent and patent schistosome infection, as such data have previously been published elsewhere by ourselves and others^{12,14,15}.*

After Figures 1 and 2 on controlled and natural human infection respectively, the authors show mouse model data from days 21 (lung stage) and 49 (patent stage). It would be preferable to show day 21 and 49 in the same figure (as is done for Fig 5) so that increases/decreases over time are more readily appraised.

- *We appreciate this point, and in previous drafts of this manuscript tried to combine d21 and d49 data (Figs 3 and 4) into a single Figure. However, we found that separating the timepoints resulted in much less cluttered, clearer figures. Having added substantial new data to both Figs 3 and 4 of the revised manuscript (40 cercariae dose, in response to Reviewers 1 and 2), we are now even more sure that combining these Figures would impair clarity and legibility of the results at each timepoint.*

Although the CD11c Δ Irf4 mice are an elegant model to demonstrate cDC2 requirements for Th2 responses in schistosomiasis, a more direct test would be to transfer DCs of different genetic and immune status to either WT or gene-targeted mice, for example to rescue the response in the Δ Irf4 mice. This would also exclude off-target effects of the transgenic constructs on other populations (for example, it is notable that the Δ Irf4 mice show a complete loss of pulmonary B cells).

- *We would suggest that experiments to try to add back defined DC subsets to the lung, or to assess the role of other antigen-presenting cells in this setting, would be technically challenging and time consuming, and outwith the scope of the current manuscript. However, we completely agree that our work as it stands cannot definitively exclude the potential role of other cell types, in addition to cDC2s, in pulmonary type-2 induction during schistosome infection, and have added new text to the Discussion to help highlight this important point (p14).*

Figure 6 A shows MGL2+ cells from WT and gene targeted mice, by implication in both cases infected; it is important to also show naive uninfected data, as the baseline levels may differ at steady state, and the degree of expansion in each genotype should be evaluated.

- *In our original submission, we did not include data from uninfected CD11c Δ Irf4 mice to try to help with clarity in the comparison between Cre+ vs Cre- infected animals. However, we agree that it is important to provide naive animal data to the reader. We have now added these data as Supp. Fig. 20, and text to p10 of the revised manuscript, to clarify that pulmonary readouts were not significantly affected by cDC2 deficiency in uninfected control animals.*

References:

1. Desch, A. N. *et al.* Flow Cytometric Analysis of Mononuclear Phagocytes in Nondiseased Human Lung and Lung-Draining Lymph Nodes. *Am J Respir Crit Care Med* **193**, 614–626 (2016).
2. Jardine, L. *et al.* Lipopolysaccharide inhalation recruits monocytes and dendritic cell subsets to the alveolar airspace. *Nat Commun* **10**, 1999 (2019).
3. Thornton, E. E. *et al.* Spatiotemporally separated antigen uptake by alveolar dendritic cells and airway presentation to T cells in the lung. *J. Exp. Med.* **209**, 1183–1199 (2012).
4. Miller, P. & Wilson, R. A. Migration of the schistosomula of *Schistosoma mansoni* from the lungs to the hepatic portal system. *Parasitology* **80**, 267–288 (1980).
5. Mayer, J. U. *et al.* Different populations of CD11b+ dendritic cells drive Th2 responses in the small intestine and colon. *Nature communications* **8**, 15820 (2017).
6. Tussiwand, R. *et al.* Klf4 expression in conventional dendritic cells is required for T helper 2 cell responses. *Immunity* **42**, 916–928 (2015).
7. Gao, Y. *et al.* Control of T helper 2 responses by transcription factor IRF4-dependent dendritic cells. *Immunity* **39**, 722–732 (2013).
8. Schlitzer, A. *et al.* IRF4 Transcription Factor-Dependent CD11b+ Dendritic Cells in Human and Mouse Control Mucosal IL-17 Cytokine Responses. *Immunity* **38**, 970–983 (2013).
9. Buckland, M. S. *et al.* Treatment of COVID-19 with remdesivir in the absence of humoral immunity: a case report. *Nat Commun* **11**, 6385 (2020).
10. Grzych, J. M. *et al.* Egg deposition is the major stimulus for the production of Th2 cytokines in murine schistosomiasis mansoni. *Journal of immunology (Baltimore, Md. : 1950)* **146**, 1322–7 (1991).
11. Pearce, E. J., Caspar, P., Grzych, J. M., Lewis, F. A. & Sher, A. Downregulation of Th1 cytokine production accompanies induction of Th2 responses by a parasitic helminth, *Schistosoma mansoni*. *Journal of Experimental Medicine* **173**, 159–166 (1991).
12. Redpath, S. A., van der Werf, N., S, M., Andrew, Maizels, R. M. & Taylor, M. D. *Schistosoma mansoni* Larvae Do Not Expand or Activate Foxp3+ Regulatory T Cells during Their Migratory Phase. *Infect. Immun.* **83**, 3881–3889 (2015).
13. Langenberg, M. C. C. *et al.* A controlled human *Schistosoma mansoni* infection model to advance novel drugs, vaccines and diagnostics. *Nature Medicine* **26**, 326–332 (2020).
14. De Oliveira Fraga, L. A., Torrero, M. N., Tocheva, A. S., Mitre, E. & Davies, S. J. Induction of type 2 responses by schistosome worms during prepatent infection. *Journal of Infectious Diseases* **201**, 464–472 (2010).
15. Costain, A. H. *et al.* Dynamics of Host Immune Response Development During *Schistosoma mansoni* Infection. *Front Immunol* **13**, 906338 (2022).

REVIEWERS' COMMENTS

Reviewer #1 (Remarks to the Author):

I thank the authors for addressing my concerns. I request the authors display the correlation between liver egg counts and BAL T cell density as a supplemental figure, as offered in the rebuttal.

Reviewer #2 (Remarks to the Author):

This reviewer appreciates the author's responses and the added data. The addition of the lower dose murine data has revealed an important dose dependency in the early inflammatory response. The addition of the discussion of the limitations in the human data is appropriate and satisfies my concerns about the interpretation of the data. It still would have been nice to assess the morbidity at later time points in the lower dose infections of the deficient mice (since they should not be sick enough to necessitate euthanasia), as the data for the low and high dose infection do suggest that there may be difference in what this response is required for in the host depending on either damage or intensity of inflammation driven by the dose. But this is not necessary for publication at this point, and the overall manuscript adds to the field and reveals critical information about the pre-patent response. The data add to previously underappreciated data in the field that have suggested a pre-patent Type 2 response, with the data in this manuscript clearly demonstrating egg-independent induction of IL-4. Overall this manuscript represents an important contribution to the field.

Reviewer #3 (Remarks to the Author):

The authors have responded to all the comments in a comprehensive and reasonable manner and I have no further issues to raise with the manuscript.

28th February 2023

Dear Reviewers,

We are pleased to submit a revised version of our manuscript entitled “**Pulmonary inflammation is a feature of human and murine schistosomiasis, promoted by type-2 dendritic cells**” for your consideration.

We thank the reviewers again for their time in considering our work and for their supportive comments and suggestions, which we feel have improved the depth and clarity of our manuscript. Below is a response to their comments. We have added one new supplementary figure as requested. Other small changes to the manuscript text are in line with editorial requests, with all changes in red.

Yours sincerely,

Andrew S. MacDonald
Professor of Immunology

Reviewer 1

I thank the authors for addressing my concerns. I request the authors display the correlation between liver egg counts and BAL T cell density as a supplemental figure, as offered in the rebuttal..

- *We thank the reviewer for their response and contributions to the manuscript. We have included the correlation between liver egg counts and BAL T cell density as a supplemental figure (Supp. Fig. 21) as requested, and have referenced this figure in the text on page 8.*

Reviewer 2

This reviewer appreciates the author's responses and the added data. The addition of the lower dose murine data has revealed an important dose dependency in the early inflammatory response. The addition of the discussion of the limitations in the human data is appropriate and satisfies my concerns about the interpretation of the data. It still would have been nice to assess the morbidity at later time points in the lower dose infections of the deficient mice (since they should not be sick enough to necessitate euthanasia), as the data for the low and high dose infection do suggest that there may be difference in what this response is required for in the host depending on either damage or intensity of inflammation driven by the dose. But this is not necessary for publication at this point, and the overall manuscript adds to the field and reveals critical information about the pre-patent response. The data add to previously underappreciated data in the field that have suggested a pre-patent Type 2 response, with the data in this manuscript clearly demonstrating egg-independent induction of IL-4. Overall this manuscript represents an important contribution to the field.

- *We thank the reviewer for their helpful comments that have improved the manuscript as well as their kind words on the importance of this contribution.*

Reviewer 3

The authors have responded to all the comments in a comprehensive and reasonable manner and I have no further issues to raise with the manuscript.

- *We thank the reviewer for their constructive comments during the revision process.*